# MTVQA: Benchmarking Multilingual Text-Centric Visual Question Answering

## Abstract

Text-Centric Visual Question Answering (TEC-VQA) in its proper format not only facilitates human-machine interaction in text-centric visual environments but also serves as a *de facto* gold proxy to evaluate AI models in the domain of text-centric scene understanding. Nonetheless, most existing TEC-VQA benchmarks focus on high-resource languages like English and Chinese. Despite pioneering works expanding multilingual QA pairs in non-text-centric VQA datasets through translation engines, the translation-based protocol encounters a substantial "visual-textual misalignment" problem when applied to TEC-VQA. Specifically, it prioritizes the text in question-answer pairs while disregarding the visual text present in images. Moreover, it fails to address complexities related to nuanced meaning, contextual distortion, language bias, and question-type diversity. In this work, we tackle multilingual TEC-VQA by introducing MTVQA, the first benchmark featuring high-quality human expert annotations across 9 diverse languages, consisting of 6,778 question-answer pairs across 2,116 images. Further, by comprehensively evaluating numerous state-of-the-art Multimodal Large Language Models (MLLMs), including GPT-4o, GPT-4V, Claude3, and Gemini, on the MTVQA dataset, it is evident that there is still a large room for performance improvement, underscoring the value of MTVQA. Additionally, we supply multilingual training data within the MTVQA dataset, demonstrating that straightforward fine-tuning with this data can substantially enhance multilingual TEC-VQA performance. We aspire that MTVQA will offer the research community fresh insights and stimulate further exploration in multilingual visual text comprehension.

## 1 Introduction

In the era of burgeoning AI, especially in MLLMs (OpenAI, 2024; Achiam et al., 2023; Yang et al., 2023; Team et al., 2023; Anthropic, 2024; Reid et al., 2024; Bai et al., 2023; Lu et al., 2024; Young et al., 2024; Feng et al., 2023a;b; Hu et al., 2024; Liu et al., 2024c; Tang et al., 2024; Chen et al., 2024; Dong et al., 2024; Li et al., 2024; Liu et al., 2024a), **T**ext-**C**entric **V**isual **Q**uestion **A**nswering (**TEC-VQA**) (Biten et al., 2019; Singh et al., 2019; Feng et al., 2023b;a; Tang et al., 2024; Liu et al., 2024c; Hu et al., 2024) has served as a *de facto* gold proxy to evaluate AI models in the domain of text-centric scene understanding. Compared with general VQA (Biten et al., 2019; Mathew et al., 2021; Pham et al., 2024; Singh et al., 2019; Mishra et al., 2019; Mathew et al., 2022; Masry et al., 2022; Zhu et al., 2016; Krishna et al., 2017; Antol et al., 2015; Marino et al., 2019; Sheng et al., 2021; Liu et al., 2024b; Gao et al., 2015; Gan et al., 2020; Liu et al., 2021), TEC-VQA places greater emphasis on answering questions that require understanding visual textual information within images. It enables individuals without specialized expertise to access applications in text-centric visual environments. However, most advancements in TEC-VQA have predominantly concentrated on high-resource languages, *e.g.*, English (Biten et al., 2019; Singh et al., 2019; Mathew et al., 2021; 2022), Chinese (Qi et al., 2022; Gao et al., 2015), Japanese (Shimizu et al., 2018; Nguyen et al., 2023) and *etc.*, thus restricting the applicability of AI models to the global community, particularly populations speaking low-resource languages.

To tackle the problem of language diversity, several seminal studies (Raj Khan et al., 2021; Pfeiffer et al., 2022; Changpinyo et al., 2023) in the general VQA field simply leverage translation engines to expand question-answer pairs from high-resource to low-resource languages. However, this

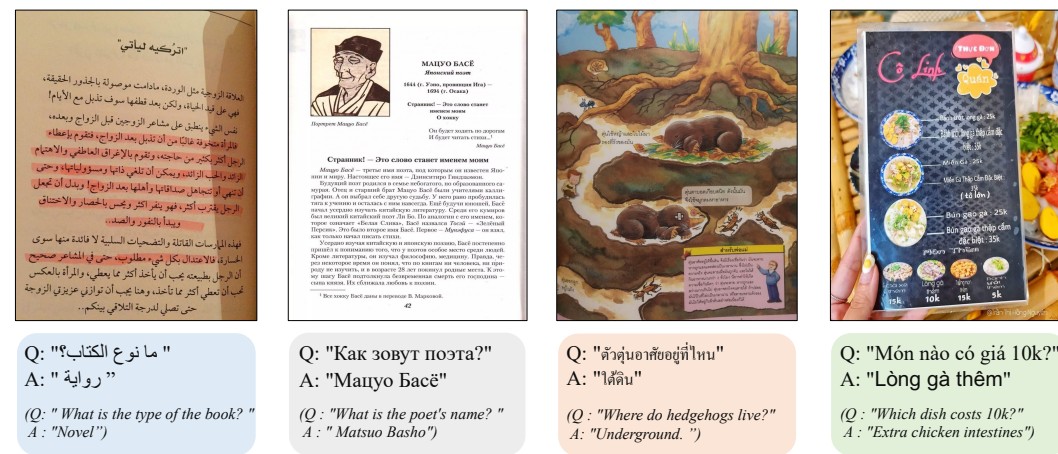

Figure 1: Multilingual text-centric VQA visualization selected from four languages. From left to right: Arabic (AR), Russian (RU), Thai (TH), Vietnamese (VI). The corresponding translations in English are in brackets. More examples see Figure 7.

translation-based approach is not feasible for TEC-VQA, as it merely processes **text in question-answer pairs**, neglecting the critical visual text needed for answering based on images. Although the visual text can be recognized by an OCR (Optical Character Recognition) engine and then translated into the target language, this indirect process could lead to a "visual-textual misalignment" problem, due to nuanced meaning, contextual distortion, language bias, and question type diversity. Taking the second case in Fig. 1 as an example, if either the visual text or the question in Russian is recognized or translated problematically, then the question would never be answered correctly. The *status quo* begs for a question: "*Can we directly leverage visual text in source language per se for multilingual TEC-VQA and what we stand in the MLLM era?*"

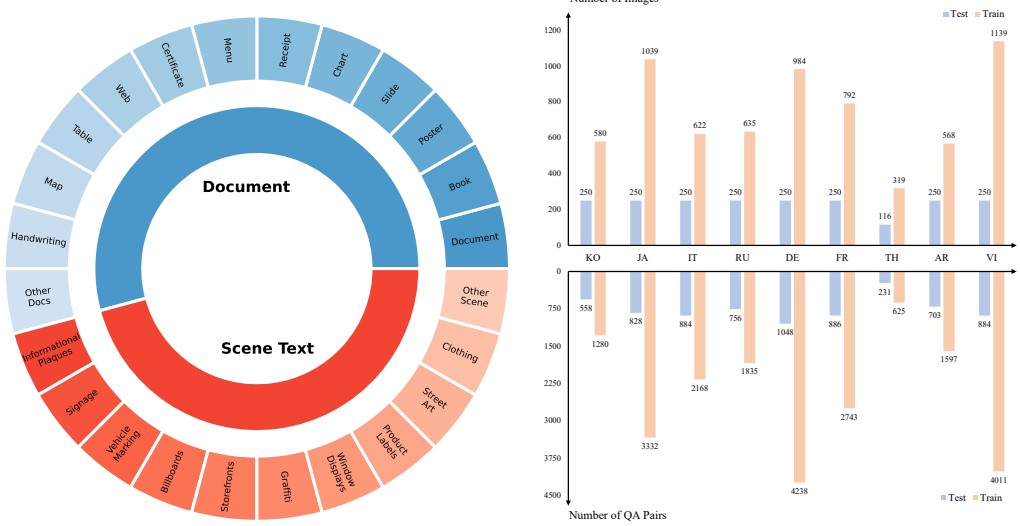

Figure 2: Left: overview of various categories of text-rich images. Right: image and QA pairs distribution over the 9 languages in MTVQA benchmark.

In this work, to answer the question above, we establish MTVQA, a novel and high-quality multilingual TEC-VQA benchmark, where all images are collected from real-world and meticulously annotated by human experts in nine languages: Arabic (AR), Korean (KO), Japanese (JA), Thai (TH), Vietnamese (VI), Russian (RU), French (FR), German (DE), and Italian (IT). More concretely, to ensure the visual-textual alignment at most, the annotation process follows the raise-then-correct paradigm, where a group of human annotators raises several distinct questions, ranging from simple

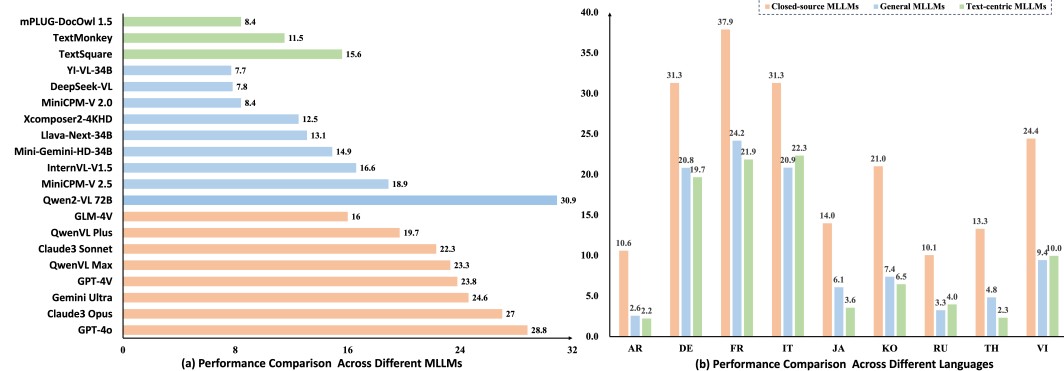

Figure 3: Left: comparison of the overall performance of various MLLMs in the MTVQA benchmark. Right: comparison of the performance exhibited by MLLMs in the 9 languages of the MTVQA.

content extraction to text-related reasoning, and subsequently provides answers. Another group then double-checks these QA pairs to ensure accuracy and consistency. Consequently, as illustrated in Fig. 2, 6,678 training images and 21,829 question-answer pairs, as well as 2,116 test images and 6,778 question-answer pairs are obtained, covering more than 20 fine-grained scenarios from both documents and natural scenes, such as menus, logos, maps, bills, PPTs, research papers, and *etc*. To our knowledge, MTVQA is the first TEC-VQA dataset to provide native human annotations for multilingual text-rich scenarios, especially for low-source languages.

We further investigate recent representative MLLMs in Fig. 3, including GPT-4o, GPT-4V, Gemini, Qwen2-VL, *etc.*, and their performance on our proposed MTVQA. As vividly revealed in Fig. 3 (a), the general MLLM Qwen2-VL is the top performer, followed by closed-source GPT-4o and Claude3 Opus. Other general MLLMs like QwenVL Max and QwenVL Plus show mid-tier performance, while text-centric MLLMs lag behind. Fig. 3(b) shows closed-source MLLMs consistently outperforming others, especially in French (FR) and German (DE), while languages like Arabic (AR) and Thai (TH) witness lower scores across all categories. The results unequivocally demonstrate that opportunities for improvement persist within existing MLLMs when applied in multilingual text-rich scenarios. In summary, the main contributions of this paper can be categorized into three points:

- We coin the MTVQA dataset, which, to the best of our knowledge, is the first multilingual TEC-VQA benchmark providing human expert annotations to solve the "visual-textual misalignment" problem in multilingual text-centric scenarios.

- We benchmark the state-of-the-art MLLMs on our new dataset and show that there are still opportunities for improvement on even the most advanced MLLMs in multilingual text-rich scenarios.

- We establish a set of new multilingual TEC-VQA baselines for closed-source and general-purpose, text-centric MLLMs.

## 2 RELATED WORK

Table 1: Comprehensive comparison on the benchmarks related to MTVQA.

| Benchmark | Scene | Manual QA | QA Language | Visual Text Language | GPT4V Performance |
|---|---|---|---|---|---|
| OCRBench | Multiple Text-rich | ✓ | English | English | 64.5% |
| TextVQA | Scene Text | ✓ | English | English | 78.0% |
| DocVQA | Document | ✓ | English | English | 88.4% |
| EST-VQA | Scene Text | ✓ | English, Chinese | English, Chinese | 72.3% |
| xGQA | General | | 7 Languages | - | 67.7% |
| MaXM | General | | 7 Languages | - | 62.8% |
| MTVQA | Multiple Text-rich | ✓ | 9 Languages | 10 languages | 22.0% |

## 2.1 MLLMs for Text-centric VQA

Recent advancements in MLLMs (Achiam et al., 2023; Yang et al., 2023; Team et al., 2023; Anthropic, 2024; Reid et al., 2024; Bai et al., 2023; Lu et al., 2024; Young et al., 2024; Feng et al., 2023a;b; Hu et al., 2024; Liu et al., 2024c; Tang et al., 2024; Chen et al., 2024; Dong et al., 2024; Li et al., 2024; Liu et al., 2024a; Zhao et al., 2024) have revolutionized VQA tasks, as demonstrated by the remarkable zero-shot performance of these models. Notably, the high generalizability of MLLMs, when explicitly trained on visual text understanding datasets and fine-tuned with instructions, has significantly enhanced their application in text-centric VQA scenarios (Feng et al., 2023b;a; Tang et al., 2024; Liu et al., 2024c; Hu et al., 2024). For example, LLaVAR (Zhang et al., 2023), UniDoc (Feng et al., 2023b), which extend LLaVA (Liu et al., 2024b) into the realm of document understanding, pioneering the text-centric VQA of MLLMs by training them to predict texts and coordinates from document images. Furthermore, DocPedia (Feng et al., 2023a) operates visual input in the frequency domain rather than in space, which enables higher input resolution without increasing the input sequence. Lately, mPLUG-DocOwl (Ye et al., 2023), Qwen-VL (Bai et al., 2023), and TextMonkey (Liu et al., 2024c) leverage publicly available document-related VQA datasets to further enhance the text-centric VQA capabilities. Despite the promising results achieved by existing MLLMs in text-centric VQA tasks, their focus on high-resource languages such as English and Chinese has posed challenges in achieving reasonable performance for low-resource languages. This is primarily due to the lack of data or benchmarks for these low-resource languages.

## 2.2 Multilingual Text-centric VQA Benchmarks

VQA has garnered significant attention in recent years, with numerous studies, datasets, and benchmarks being proposed to advance the field (Biten et al., 2019; Mathew et al., 2021; Pham et al., 2024; Singh et al., 2019; Mishra et al., 2019; Mathew et al., 2022; Masry et al., 2022; Zhu et al., 2016; Krishna et al., 2017; Antol et al., 2015; Marino et al., 2019; Sheng et al., 2021; Liu et al., 2024b; Gao et al., 2015; Gan et al., 2020; Liu et al., 2021). Many datasets have been created that encompass scene text of various domains, including natural images (Biten et al., 2019; Singh et al., 2019), scanned documents (Mathew et al., 2021; 2022), book and movie covers (Mishra et al., 2019). One notable limitation of these datasets is their predominant focus on English (Biten et al., 2019; Singh et al., 2019; Mathew et al., 2021; 2022) or other high-resource languages such as Chinese (Qi et al., 2022; Gao et al., 2015) and Japanese (Shimizu et al., 2018; Nguyen et al., 2023), which restricts the applicability of VQA systems for low-resource languages such as Thai and Vietnamese.

There are some recent efforts toward extending VQA tasks to a broader range of languages (Gupta et al., 2020; Pfeiffer et al., 2022; Vivoli et al., 2022; Changpinyo et al., 2023; Li et al., 2023; Raj Khan et al., 2021) by providing a multilingual VQA datasets. For example, Gao et al. (2015) created a free-form bilingual VQA dataset (FM-IQA) containing over 150,000 images and 310,000 freestyle Chinese question-answer pairs and English translations. Raj Khan et al. (2021) developed a large-scale multilingual and code-mixed VQA dataset (MuCo-VQA) supporting five languages. Of more relevance are the works xGQA (7 languages) (Pfeiffer et al., 2022) and MaXM (7 languages) (Changpinyo et al., 2023), which apply translation-based protocols to expand VQA data beyond English. However, the translation-based multilingual VQA datasets inherently face issues, such as the "visual-textual misalignment" problem, where only the textual information in question-answer pairs is considered, while the visual text in images is overlooked. Additionally, the nuanced meaning and context are often distorted; language bias is introduced by machine translation models, and the coverage of certain question types is limited, as highlighted by Changpinyo et al. (2023). Moreover, none of the previous multilingual datasets focus on text-centric scenarios where multilingual text frequently occurs.

Our benchmark, MTVQA, distinguishes itself by focusing on multilingual text-centric VQA scenarios using human expert annotations. It covers 9 languages, facilitating the training and evaluation of multilingual models in diverse linguistic contexts. Additionally, our dataset can gauge the VQA system's ability for not only high-resource languages but also those that are typically underrepresented in current datasets (Biten et al., 2019; Singh et al., 2019; Mathew et al., 2021; 2022; Gao et al., 2015).

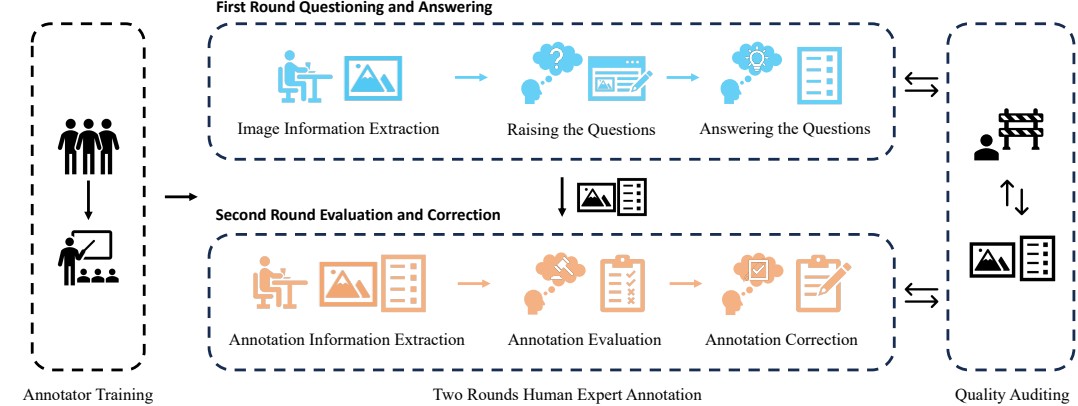

Figure 4: A brief diagram of the annotation process.

# 3 MTVQA BENCHMARK

The MTVQA benchmark is meticulously established to evaluate the multilingual text comprehension performance of Multimodal Large Language Models (MLLMs). MTVQA covers nine languages: Arabic (AR), Korean (KO), Japanese (JA), Thai (TH), Vietnamese (VI), Russian (RU), French (FR), German (DE), and Italian (IT). The data construction involves a comprehensive collection of text-rich images and a two-round human expert annotation process. In terms of data volume, an initial dataset of 8,980 images is compiled. Following a data cleaning phase, 8,895 images are retained for annotation. The first round of annotation yielded 8,895 images, corresponding to 28,906 question-answer (QA) pairs. After the second round of annotation and final quality control measures, the final dataset comprises 8,794 images and 28,607 QA pairs. The benchmark construction cost is divided into two parts: image acquisition and annotation. Image acquisition costs about two months and 30,000 dollars. VQA annotation costs about three months and 60,000 dollars. Hourly wages vary from country to country, with senior language specialists costing between 20 and 40 per hour and local annotators costing between 8 and 20 per hour. The average time spent by each labeler was roughly 60 days.

## 3.1 DATA COLLECTION

The raw data collection aims to gather text-rich images from various scene text and document scenarios, ensuring diversity and quality. This includes images from publicly available datasets (e.g., ICDAR MLT19 (Gao et al., 2019)) and those sourced from the internet (i.e., Laion-OCR, which is filtered from Common Crawl (Crawl, 2024)), such as menus, logos, maps, bills, PowerPoint slides (PPTs), research papers, etc, as shown in Fig. 2. The image collection process consists of two steps: extracting multilingual text using a multilingual OCR engine and selecting by language types and the amount of text contained in the image. Approximately 30% of the overall image data is obtained from public datasets, with 20% sourced from the web and 50% from manual collection, respectively. A total of 1,220 images from document scenarios and 876 images from natural scenarios are collected for the test set of the MTVQA benchmark. To ensure data content meets regulatory requirements, we subject them to a standardized data cleaning process. The image-cleaning processing pipeline involves a preliminary round of algorithm-driven review to identify and filter out unusable images. It includes detecting and removing images with politically sensitive, pornographic, violent, or other undesirable features. Subsequently, a multilingual OCR tool is employed to extract the textual content from the remaining images. Images devoid of textual information are discarded, and the surviving images are categorized based on their language. Afterward, we organize all the text-rich images we have obtained into language-specific groups, preparing them for the subsequent stage of data annotation.

## 3.2 Human Expert Annotation

To obtain informative and accurate text-related QA pairs for images grouped by specific languages, a specialized group of annotators with expertise in the local regions of each language is recruited. The annotators must be native speakers of the corresponding language and have actively utilized it for a minimum duration of 10 years. Additionally, they must possess at least a university-level degree or higher academic qualification, guaranteeing a profound understanding and skillfulness in the linguistic subtleties and cultural contexts required for precise annotations. Given the subjective nature of understanding text within images, the annotation team is divided into two independent groups. One group is tasked with generating questions and providing answers based on the images, while the other group is responsible for evaluating and correcting the QA pairs. This division, known as the raise-then-correct paradigm, ensures a thorough and trustworthy evaluation of the text-rich image comprehension process. Moreover, the annotation results for each language underwent a 10% sampling inspection by a quality inspector to ensure adherence to standards. QA pairs that do not meet the criteria are returned for re-annotation. Prior to the formal annotation process, all annotators will be provided with a detailed explanation of the annotation guidelines, including a dedicated question-and-answer session to clarify any ambiguities and uncertainties. A pilot annotation task will be conducted on a limited dataset to ensure a shared understanding of the annotation rules. The two-round annotation process is briefly illustrated in Fig. 4, further details of which are elaborated in the subsequent sections.

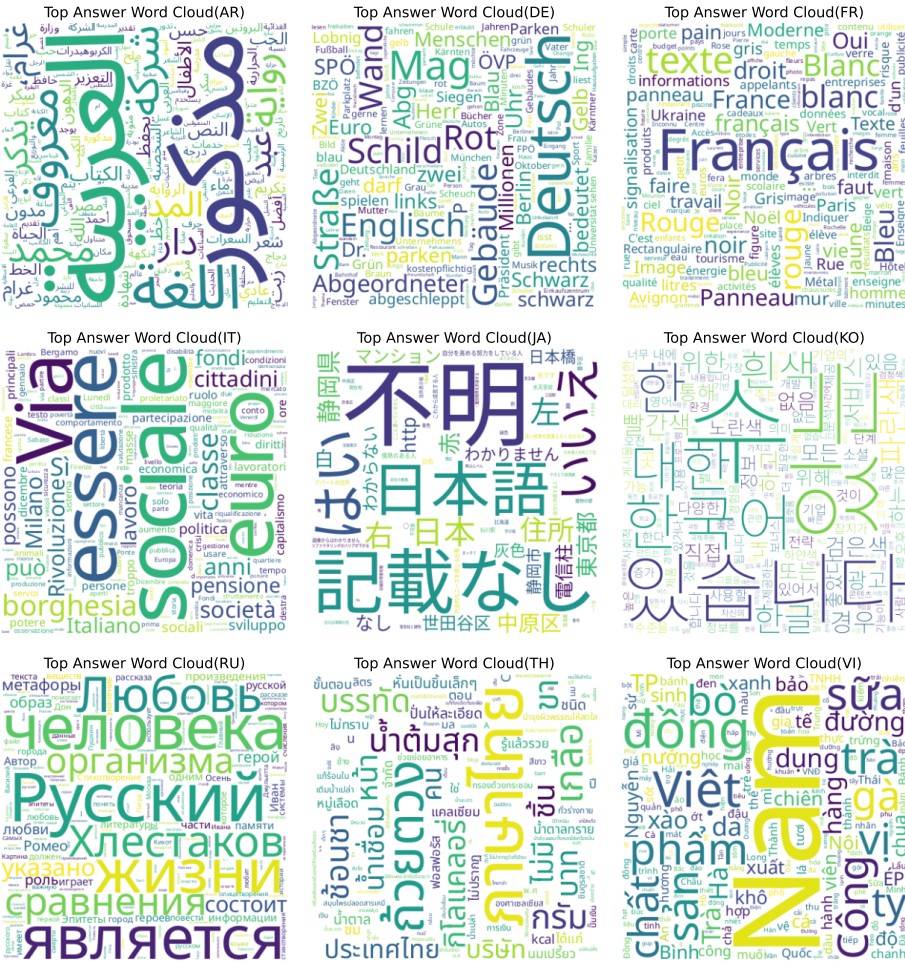

Figure 5: Word clouds showcasing top answers in various languages, tokenized via NLTK with removing stop words, punctuation, and digits.

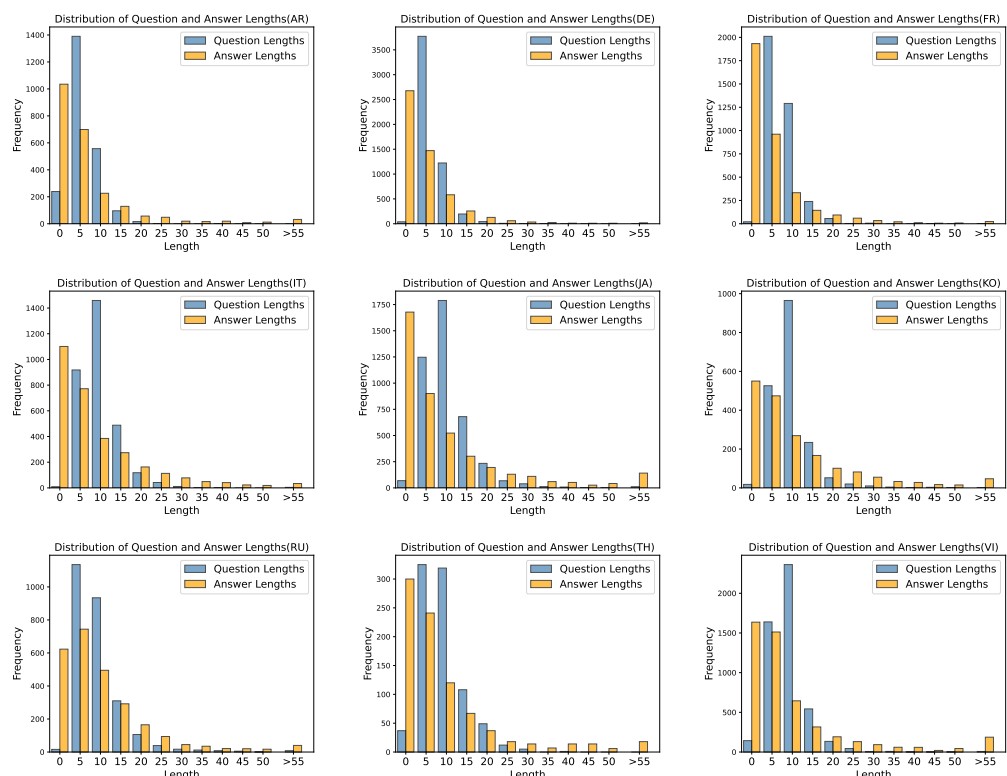

Figure 6: Statistics of question and answer lengths of different languages aggregating training and test set, using GPT-4o tokenizer.

**First Round Questioning and Answering.** In the first round, we allocate three annotators per language to generate initial QA results. The annotators are tasked with thoroughly examining the textual and visual elements within a text-rich image in our collection. They are to examine the textual and visual elements within the image to formulate five meaningful and distinct questions and provide corresponding answers. All annotators should adhere to the following criteria: (1) the first three questions should satisfy that answering these questions requires direct reading of the textual information in the image, (2) the fourth and fifth questions require reasoning about the text in the image to answer, (3) the questions and answers must be reasonably correct and consistent with the content of the image, and (4) the answer should be as concise as possible and free of nonsense (*e.g.*, when the question is "When is the volunteer recruitment period", the answer should be "9:00-16:00" rather than "The volunteer recruitment period is 9:00-16:00"). This requirement for brevity aims to make the evaluation process user-friendly and reliable, avoiding the influence of extraneous content on the evaluation metrics.

**Second Round Evaluation and Correction.** To reduce the effect of human subjective cognitive bias on our MTVQA benchmark and get high-quality question-answer pairs, we assign two annotators for each language to carry out the annotation evaluation and correction process independently. These annotators follow a specific evaluation and correction protocol to ensure consistency and accuracy based on the first round results: (1) Assess the question's relevance to the text in the image. Irrelevant QA pairs are discarded; (2) Verify the correctness of the answer and make necessary modifications; (3) Check for redundancy in the answer. If the answer repeats the question's information, the repeated content is removed for conciseness. (4) Ethical Assessment. Review the image content or the question-answer pair for any unethical content, including but not limited to politics, personal privacy issues, etc. Such content is removed to uphold ethical standards in the dataset.

3.3 DATA STATISTICS

The MTVQA benchmark consists of 8,794 images and 28,607 question-answer pairs across the nine languages, divided into a training set with 6,678 images and 21,829 question-answer pairs and a test set with 2,116 images and 6,778 question-answer pairs. Detailed data distribution is illustrated in Fig. 2. Additionally, the benchmark showcases the vocabulary richness for each language through word clouds, as depicted in Fig. 5, and the lengths of questions and answers are statistically analyzed using the GPT-4o tokenizer, as shown in Fig. 6.

4 EXPERIMENTS

4.1 BASELINE MODELS

To comprehensively assess MLLMs' multilingual perception and comprehension capabilities, We select state-of-the-art MLLMs from three categories: open-source general MLLMs, open-source text-centric MLLMs, and closed-source MLLMs. Each category contains the following models: (1) **General MLLMs:** Qwen2-VL  Wang et al. (2024), InternVL-V1.5 (Chen et al., 2023), InternLM-Xcomposer2-4KHD (Dong et al., 2024), Mini-Gemini-HD-34B (Li et al., 2024), Llava-Next-34B (Liu et al., 2024a), DeepSeek-VL (Lu et al., 2024), YI-VL-34B (Young et al., 2024); (2) **Text-centric MLLMs:** TextSquare (Tang et al., 2024), TextMonkey (Liu et al., 2024c), mPLUG-DocOwl 1.5 (Hu et al., 2024), MiniCPM-V 2.0 (Hu et al., 2023); (3) **Closed-source MLLMs:** GPT-4o (OpenAI, 2024), GPT-4V (Achiam et al., 2023), Gemini Ultra (Team et al., 2023), QwenVL Max (Bai et al., 2023), QwenVL Plus (Bai et al., 2023), Claude3 Opus (Anthropic, 2024), Claude3 Sonnet (Anthropic, 2024), and GLM-4V (AI, 2024). For the closed-source MLLMs, we use the chat version through official APIs, while for the open-source MLLMs, we utilize the instruct versions available on the HuggingFace Model Hub. The open-source MLLMs' model size varies from 7b to 76b.

4.2 IMPLEMENTATION DETAILS

We conduct the evaluation experiments over the baseline MLLMs with their default settings, ignoring the effect of generation configuration on the results. To make the output of MLLMs more evaluation-friendly, we design the following prompt format to limit the output length: "Answer the question using a word or phrase in the language of the question. + <Question>", where "<Question>" represents the actual question from the MTVQA test set. This approach aims to make the answers as concise as possible. Besides, InternLM-Xcomposer2-4KHD  (Dong et al., 2024) is chosen as the base model for an instruction tuning experiment on the MTVQA training set. The instruction tuning process adheres to the default training settings specified by the source, with "HD-16" and completes one epoch of training on 8 NVIDIA-A100 GPUs within 2 hours.

4.3 EVALUATION RESULTS

**Evaluation metric.** To accurately assess whether the visual text that occurs in the answer is correct, we adopt Accuracy as the metric. The Accuracy metric measures the percentage of questions for which the predicted answer matches exactly with any of the target answers for the question. The accuracy metric awards a zero score even when the prediction differs slightly from the target answer.

**Zero-shot evaluation.** We perform a zero-shot evaluation of various types of MLLMs on the MTVQA benchmark with a consistent prompt. The evaluation results are shown in Table 2, where Qwen2VL 72B (Wang et al., 2024) achieves the highest average accuracy of 30.9% and GPT-4o (OpenAI, 2024) achieves the second highest average accuracy of 27.8% across the 9 languages. This result suggests that while MLLMs have some capability in comprehending multilingual text, the performance is still not robust, and multilingual text-centric visual question answering (VQA) tasks remain a significant challenge, even for state-of-the-art closed-source MLLMs. The evaluation also reveals that both open-source and closed-source models performed better on Indo-European languages that use the Latin alphabet, such as German (DE), French (FR), and Italian (IT). This is likely due to the more extensive training data available for English and their visual and linguistic similarities. In addition, most closed-source models outperform the open-source models except for Qwen2VL across the nine languages, potentially benefiting from pre-training on diverse, multilingual data. Interestingly, the text-centric MLLMs, like TextSquare (Tang et al., 2024), TextMonkey (Liu et al., 2024c) and

Table 2: Performance of the leading closed- and open-source MLLMs on the MTVQA benchmark. The best results of each language are **bolded**. The second best results are underlined. "Xcomposer-SFT" denotes instruction tuning to Xcomposer2-4KHD with MTVQA's training set.

| | AR | DE | FR | IT | JA | KO | RU | TH | VI | Avg. |
|---|---|---|---|---|---|---|---|---|---|---|
| *Closed-Source MLLMs* | | | | | | | | | | |
| GPT-4V (Achiam et al., 2023) | 11.5 | 31.5 | 40.4 | 32.3 | 11.5 | 16.7 | 10.3 | 15.0 | 28.9 | 22.0 |
| GPT-4o (OpenAI, 2024) | 20.2 | 34.2 | 41.2 | 32.7 | **20.0** | **33.9** | 11.5 | **22.5** | 34.2 | 27.8 |
| Gemini Ultra (Team et al., 2023) | 14.7 | 32.3 | 40.0 | 31.8 | 12.3 | 17.2 | 11.8 | 20.3 | 28.6 | 23.2 |
| QwenVL Max (Bai et al., 2023) | 7.7 | 31.4 | 37.6 | 30.2 | 18.6 | 25.4 | 10.4 | 4.8 | 23.5 | 21.1 |
| QwenVL Plus (Bai et al., 2023) | 4.8 | 28.8 | 33.7 | 27.1 | 12.8 | 19.9 | 9.4 | 5.6 | 18.1 | 17.8 |
| Claude3 Opus (Anthropic, 2024) | 15.1 | 33.4 | 40.6 | 34.4 | 19.4 | 27.2 | 13.0 | 19.5 | 29.1 | 25.7 |
| Claude3 Sonnet (Anthropic, 2024) | 10.5 | 28.9 | 35.6 | 31.8 | 13.9 | 22.2 | 11.0 | 15.2 | 20.8 | 21.1 |
| GLM-4V (AI, 2024) | 0.3 | 30.0 | 34.1 | 30.1 | 3.4 | 5.7 | 3.0 | 3.5 | 12.3 | 13.6 |
| *Open-Source MLLMs* | | | | | | | | | | |
| Qwen2-VL 72B (Wang et al., 2024) | **20.7** | **36.5** | **44.1** | **42.8** | **21.6** | **37.4** | **15.6** | 17.7 | **41.6** | **30.9** |
| InternVL2 76B (Achiam et al., 2023) | 9.5 | 31.3 | 35.7 | 35.2 | 11.1 | 14.3 | 11.9 | 10.0 | 26.9 | 22.0 |
| InternVL-V1.5 (Chen et al., 2023) | 3.4 | 27.1 | 31.4 | 27.1 | 9.9 | 9.0 | 4.9 | 8.7 | 12.4 | 14.9 |
| Mini-Gemini-HD-34B (Li et al., 2024) | 2.2 | 25.0 | 29.2 | 25.5 | 6.1 | 8.6 | 4.1 | 4.3 | 11.8 | 13.0 |
| Llava-Next-34B (Liu et al., 2024a) | 3.3 | 24.0 | 28.0 | 22.3 | 3.6 | 6.1 | 2.6 | 0.4 | 9.8 | 11.1 |
| DeepSeek-VL (Lu et al., 2024) | 0.6 | 14.2 | 15.3 | 15.2 | 2.9 | 3.8 | 1.6 | 0.9 | 5.2 | 6.6 |
| YI-VL-34B (Young et al., 2024) | 1.7 | 13.5 | 15.7 | 12.1 | 4.8 | 5.2 | 0.8 | 3.5 | 4.1 | 6.8 |
| MiniCPM-V 2.0 (Hu et al., 2023) | 1.3 | 12.7 | 14.9 | 17.0 | 3.7 | 5.6 | 2.2 | 2.2 | 6.8 | 7.4 |
| MiniCPM-V 2.5 (Hu et al., 2023) | 6.1 | 29.6 | 35.7 | 26.0 | 12.1 | 13.1 | 5.3 | 12.6 | 15.3 | 17.3 |
| TextSquare (Tang et al., 2024) | 3.7 | 27.0 | 30.8 | 26.7 | 3.2 | 7.2 | 6.7 | 5.2 | 12.4 | 13.6 |
| TextMonkey (Liu et al., 2024c) | 2.0 | 18.1 | 19.9 | 22.1 | 4.6 | 7.2 | 3.2 | 0.9 | 11.1 | 9.9 |
| mPLUG-DocOwl 1.5 (Hu et al., 2024) | 1.0 | 13.9 | 14.9 | 18.2 | 2.9 | 5.0 | 2.0 | 0.9 | 6.4 | 7.2 |
| Xcomposer2-4KHD (Dong et al., 2024) | 2.0 | 20.6 | 23.2 | 21.6 | 5.6 | 7.7 | 4.1 | 6.1 | 10.1 | 11.2 |
| Xcomposer-SFT | 11.8 | 31.7 | 37.4 | 29.3 | 14.5 | 12.9 | 5.8 | 13.9 | 20.2 | 19.7 |

Table 3: Few-shot performance of GPT-4V on the MTVQA benchmark. In-context examples are randomly selected from the training set of MTVQA in the respective languages. "n-shot" represents the number of the selected in-context examples.

| | AR | DE | FR | IT | JA | KO | RU | TH | VI | Avg. |
|---|---|---|---|---|---|---|---|---|---|---|
| zero-shot | 11.5 | 31.5 | 40.4 | 32.3 | 11.5 | 16.7 | 10.3 | 15.0 | 28.9 | 22.0 |
| two-shot | 11.6 | **35.3** | 42.2 | 33.1 | 13.2 | 17.0 | 11.4 | 19.0 | 31.0 | 23.7 |
| five-shot | 13.5 | 34.8 | **43.0** | **35.7** | **13.4** | **18.8** | 11.6 | **19.2** | **33.0** | **24.8** |
| eight-shot | **15.5** | 34.8 | 42.1 | 35.6 | **13.4** | 17.5 | **12.1** | 18.2 | 32.8 | 24.7 |

mPLUG-DocOwl 1.5 (Hu et al., 2024), do not show a significant performance advantage over other open-source models for the languages tested, suggesting a focus on high-resource languages (mainly English and Chinese) and a lack of attention to other languages.

**Instruction tuning.** As shown in Table 2, the instruction tuning experiment on MTVQA benchmark brings a 8.5% improvement. Concerning specific languages, French sees the largest improvement of 14.2% in accuracy, while Russian has the smallest improvement of 1.7%. The results demonstrate that MLLMs vary in their ability to understand and learn from text-centric data in different languages, leaving great potential for future research of multilingual text-centric MLLMs pre-training.

**Few-shot evaluation of GPT-4V.** Here, we compare the performance of GPT-4V on MTVQA under the few-shot settings. Specifically, we perform zero-shot, two-shot, five-shot, and eight-shot evaluations. We randomly select in-context examples from the train set in the respective languages and evaluate GPT-4V on the remaining instances. As shown in Tab. 3, compared to the zero-shot setting, GPT-4V's performance has improved considerably under few-shot, highlighting its exceptional in-context learning ability in multilingual text comprehension contexts. Moreover, comparisons based on varying numbers of in-context samples reveal that augmenting the in-context samples can aid in further enhancement; however, after reaching a certain volume, the improvement becomes saturated.

Table 4: Comparison on applying OCR to GPT-4 and GPT-4V

|            | AR   | DE   | FR   | IT   | JA   | KO   | RU   | TH   | VI   | Avg. |
|------------|------|------|------|------|------|------|------|------|------|------|
| OCR+GPT-4  | 18.9 | 21.2 | 29.8 | 27.3 | 15.8 | 24   | 10.6 | 19   | 28.2 | 21.6 |
| OCR+GPT-4V | 22.3 | 35.1 | 42.6 | 36.7 | 19.2 | 32.1 | 12.1 | 20.9 | 34.1 | 28.3 |
| GPT-4V     | 11.5 | 31.5 | 40.4 | 32.3 | 11.5 | 16.7 | 10.3 | 15   | 28.9 | 22.0 |

**Experiments on Text-Based LLMs and MLLMs with OCR results.** To compare text-based LLMs and MLLMs performance on the MTVQA bench with OCR results, we adopt the Bytedance multilingual OCR API tool to extract multilingual text from images. For text-based LLMs, we choose GPT-4, and for MLLMs, we choose GPT-4V. As shown in Table 4, GPT-4V and OCR+GPT-4 have similar performance, but both are much lower than OCR+GPT-4V. We analyze the cases in detail and find differences in the challenges encountered by OCR+GPT-4 and GPT-4V. Since the performance of Latin languages is weaker than that of GPT4V(strong Latin text perception and comprehension abilities), OCR+GPT4 issues lie more in the lack of perception of visual elements and position. Many questions are dependent on visual elements and positional relationships in the image, resulting in that cannot be answered. GPT-4V issues lie more in the weakness of perception and comprehension of visual text, especially non-Latin text (AR/JA/KO/TH in Table 4), but GPT-4V can capture visual elements and positional relationships in images. More importantly, all three settings perform poorly and the comprehension of multilingual visual text remains a challenging case.

## 5 LIMITATION

The current version of the MTVQA dataset, while dialectically diverse, exhibits limitations in its language coverage. Despite encompassing a range of languages, it falls short of inclusivity, omitting numerous lesser-spoken languages. This prompts our future continual endeavor to ensure comprehensive representation across the linguistic spectrum. Additionally, the dataset currently provides a single canonical response per question, which may not fully capture the range of answers corresponding to different expressions of the same underlying semantics. Recognizing the multifaceted nature of the inquiries, future versions will aim to include a spectrum of plausible answers to reflect the varied perspectives inherent to each question.

## 6 CONCLUSION

To attack the visual-textual misalignment issue in multilingual TEC-VQA, we introduce MTVQA, a new benchmark featuring high-quality human expert annotations in 9 diverse languages. We believe that MTVQA is the first benchmark to provide fully manual annotations tailored to text-centric scenarios. The results obtained from closed-source, general-purpose, and text-centric MLLMs on our MTVQA dataset indicate that there is still room for improving their performance in multilingual text-centric scenarios. Although the MTVQA dataset has limitations, such as the underrepresentation of several lesser-spoken languages and single canonical answers, future updates will address these issues by expanding the multilingual scope and including a range of plausible answers. We are confident that this dataset can inspire researchers within the TEC-VQA community with new perspectives and ideas.

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

## A    MORE VISUALIZATIONS

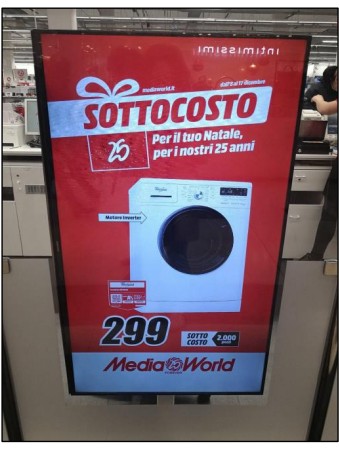

Q: "Wer ist Nicole Geipel?"
A: "Eine glückliche Single und studiert Human Resources."

*(Q: "Who is Nicole Geipel?"*
*A : "A happy single who is studying Human Resources.")*

(a) German (DE) example

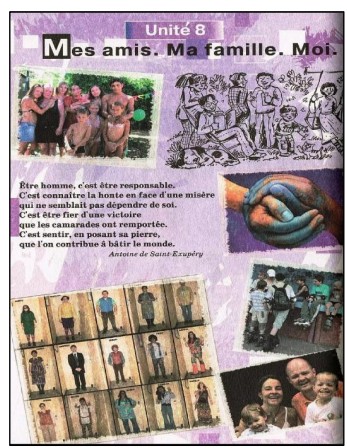

Q: "Qual è il prezzo della lavatrice pubblicizzata?"
A: "299 euro"

*(Q: "What is the price of the advertised washing machine?"*
*A : "299 euros")*

(b) Italian (IT) example

Q: "Qui est responsable ?"
A: "Être homme, c'est être responsable."

*(Q: "Who is responsible?"*
*A : "To be human is to be responsible.")*

(c) French (FR) example

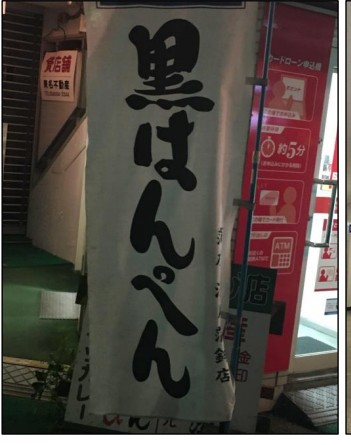

Q: "この写真の商品は何ですか？ "
A: "黒はんぺん"

*(Q: "What is the product in this picture?"*
*A : "Black hanpen")*

(d) Japanese (JA) example

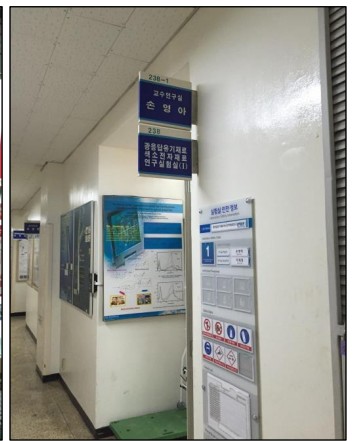

Q: "238-1에 쓰여있는 내용은?"
A: "교수연구실 손영아"

*(Q: "What is written on 238-1?"*
*A : "Professor's office, Son Young-ah.")*

(e) Korean (KO) example

Figure 7: Multilingual VQA examples selected from five languages. The corresponding translations in English are in brackets.

Table 5: Mean lengths of question-answer pairs in different languages of the training set and test set, using GPT-4o tokenizer.

| | AR | DE | FR | IT | JA | KO | RU | TH | VI |
|---|---|---|---|---|---|---|---|---|---|
| *Training Set* | | | | | | | | | |
| Question | 8.29 | 8.72 | 9.73 | 12.05 | 12.43 | 11.74 | 11.56 | 11.35 | 11.21 |
| Answer | 9.66 | 6.96 | 7.34 | 11.24 | 12.70 | 13.56 | 12.00 | 11.26 | 13.31 |
| *Test Set* | | | | | | | | | |
| Question | 8.08 | 8.29 | 9.76 | 11.93 | 12.48 | 12.2 | 11.65 | 10.98 | 10.99 |
| Answer | 7.95 | 6.67 | 6.61 | 11.04 | 12.55 | 13.61 | 14.42 | 12.08 | 13.00 |

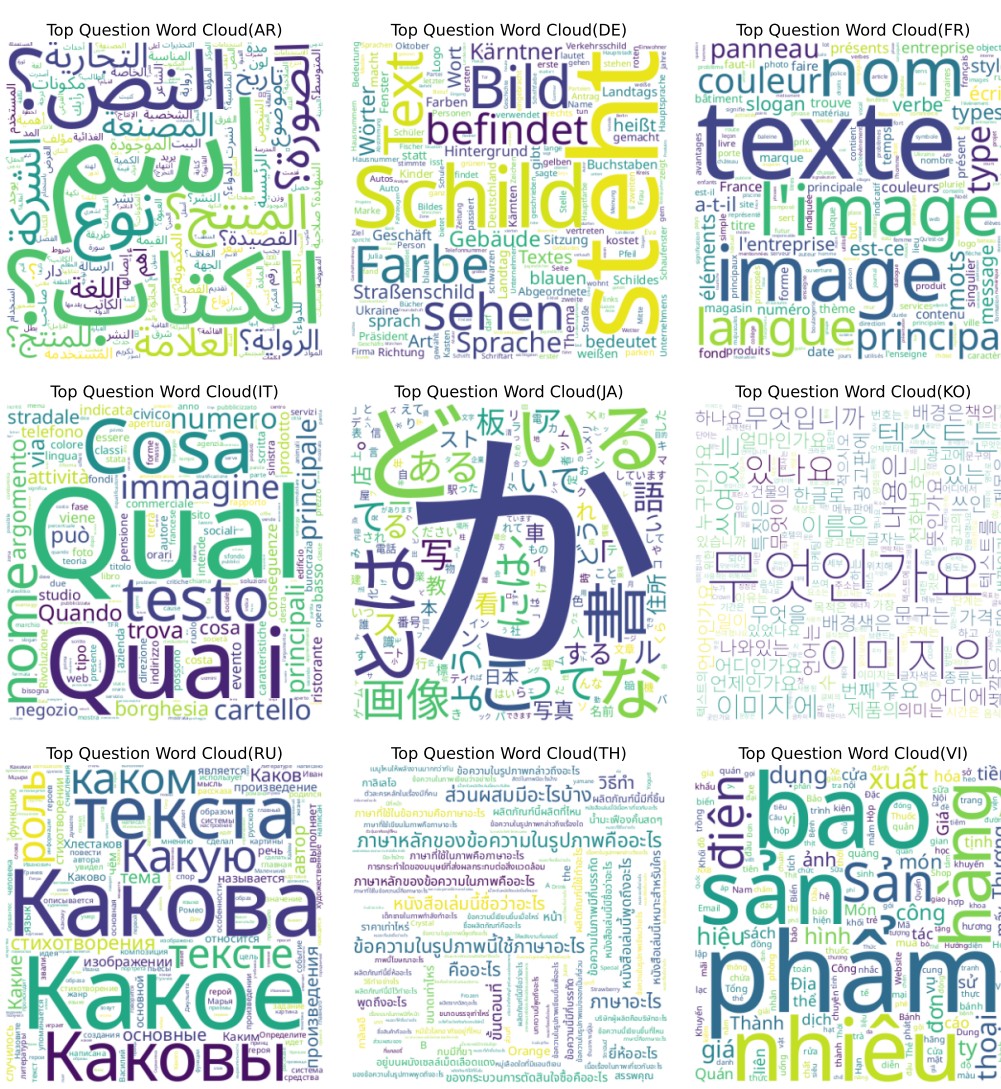

Figure 8: Word clouds showcasing top questions in various languages, tokenized via NLTK with the removal of stop words, punctuation, and digits.

# B MORE EXPERIMENTS

## B.1 MLLM PERFORMANCE CHANGES WHEN THE QUESTION IS ASKED IN ENGLISH.

We translate the questions into English using GPT-4 and then perform a test on GPT4V. The result is shown in Table 6. The GPT4V performance is robust when the question is asked in English and the original languages.

Table 6: GPT4V performance changes when the question asked in English

|  | AR | DE | FR | IT | JA | KO | RU | TH | VI | Average |
|---|---|---|---|---|---|---|---|---|---|---|
| English Questions | 11.9 | 31.8 | 39.0 | 32.1 | 12.0 | 15.8 | 10.1 | 18.1 | 28.5 | 22.1 |
| Original Questions | 11.5 | 31.5 | 40.4 | 32.3 | 11.5 | 16.7 | 10.3 | 15.0 | 28.9 | 22.0 |

