# OpenReview forum: "MTVQA: Benchmarking Multilingual Text-Centric Visual Question Answering"
_ICLR.cc/2025/Conference — ICLR 2025 Conference Withdrawn Submission_

### Official Review · Reviewer_Ar3R · 2024-10-16

**Soundness:** 2
**Presentation:** 3
**Contribution:** 2
**Rating:** 3
**Confidence:** 4

**Summary:**

This paper presents a dataset for text-based VQA, containing 8,794 images and 28,607 QA pairs. The production cost is about 90K USD. The dataset contains seven languages, including Arabic (AR), Korean (KO), Japanese (JA), Thai (TH), Vietnamese (VI), Russian (RU), French (FR), German (DE), and Italian (IT). The paper tested a number of models under zero-shot settings, GPT-4V under in-context-learning few-shot, and GPT-4 with OCR.

**Strengths:**

- The dataset is reasonably large, though the number of images is still limited and there is only one answer per question.
- The zero-shot evaluation used a large number of models.
- The tested models perform poorly on this dataset, suggesting that the dataset could have value.

**Weaknesses:**

- The exact match error metric is overly restrictive, which is partially why the tested models do not achieve high scores. Even two humans would not answer the same question exactly the same. The authors should introduce a multiple-choice metric. In addition, they should establish a human baseline, where human annotators are asked to answer the same questions. This would serve two purposes: (1) it validates the provided answers and (2) it shows where the ceiling of performance is. Given the metric, it is possible that even humans cannot guess the exact answer.

- In Line 74, the authors claim that translation does not work for TEC-VQA, since it would ignore the text in the image. However, this was never tested. The author should have a baseline where only the question is translated, and a baseline that performs OCR and translation.

- The dataset is small relative to the development cost. In particular, the authors should explain why image acquisition cost about 30,000 dollars and what kind of manual collection was used.

Minor comments:

Line 79 reads: "The status quo begs for a question: 'Can we directly leverage visual text in source language per se for multilingual
TEC-VQA and what we stand in the MLLM era?'" But it's two questions.

Can you explain the content of "regulatory requirements" (Line 263)? In particular, what is considered "politically sensitive"?

**Questions:**

See weaknesses

---

### Official Review · Reviewer_iR1C · 2024-10-31

**Soundness:** 2
**Presentation:** 3
**Contribution:** 2
**Rating:** 5
**Confidence:** 5

**Summary:**

The paper constructs the multilingual TEC-VQA benchmark, where all images are collected from real-world and  annotated by human experts in nine languages: Arabic (AR), Korean (KO), Japanese (JA), Thai (TH), Vietnamese (VI), Russian (RU), French (FR), German (DE), and Italian (IT). They evaluate numerous MLLMs on the MTVQA dataset. Additionally, we supply multilingual training data within the MTVQA dataset.

**Strengths:**

1. All images are collected from real-world and annotated by human experts in nine languages. The benchmark consists of 6,678 training images and 21,829 question-answer pairs, as well as 2,116 test images and 6,778 question-answer pairs.
2. By evaluating numerous state-of-the-art MLLMs, including GPT-4o, GPT-4V, Claude3, and Gemini, on the MTVQA dataset, it is evident that there is still a large room for performance improvement, underscoring the value of MTVQA.

**Weaknesses:**

1. The amount of data is not substantial，can it sufficiently evaluate the differences among various MLLMs across different languages and domains? There is a lack of more detailed error analysis.
2. Lacking some more insightful findings, such as the factors influencing multilingual capabilities, the difficulty and generalizability of different languages, etc.
3. The improvement of  incontext in Table 3 is not very significant, what could be the reason for the considerable differences across different languages?

**Questions:**

see weaknesses.

---

### Official Review · Reviewer_Lqka · 2024-11-01

**Soundness:** 3
**Presentation:** 3
**Contribution:** 3
**Rating:** 6
**Confidence:** 4

**Summary:**

This paper proposes MTVQA dataset ,which is claimed as the first Text-Centric Visual Question Answering (TEC-VQA) benchmark providing human expert annotations to solve the “visual-textual misalignment” problem in multilingual text-centric scenarios. Experimental results show that existing Multimodal Large Language Models remains a large room for improvement.

**Strengths:**

1. The human expert annotations presented in this paper is valuable.

2. This paper empirically analyze the multilingual VQA abilities of current leading MLLMs on their benchmark.

**Weaknesses:**

1. The results presented in Appendix B.1 show that GPT-4o struggles to understand certain languages. It would be helpful to provide results for some MLLMs when given ground truth English questions, rather than questions translated by the model itself.

2. Since the SoTA performance on this benchmark is only around 30% accuracy, it is important to include human performance to demonstrate that the questions in this benchmark are not ill-defined.

**Questions:**

Please see weakness.

---

### Official Review · Reviewer_eVLL · 2024-11-03

**Soundness:** 3
**Presentation:** 2
**Contribution:** 2
**Rating:** 5
**Confidence:** 4

**Summary:**

This paper provides a Text-Centric Visual Question Answering (TEC-VQA) benchmark consisting of nine languages: Arabic (AR), Korean (KO), Japanese (JA), Thai (TH), Vietnamese (VI ), Russian (RU), French (FR), German (DE), Italian (IT) texts in images, and questions and answers about those images in a raise-then-correct paradigm, resulting in over 6000 QA pairs for over 2000 images. The results of the exact match evaluation show that the accuracy is at most 30% for closed-source MLLMs and open-source MLLMs, indicating that this benchmark is challenging.

**Strengths:**

- Release of datasets that allow benchmarking as a combination of multimodal and multilingual tasks
- Evaluation experiments with many MLLMs

**Weaknesses:**

- Benchmark data quality
  - For example, looking at the third example in Figure 1, the question “Where do hedgehogs live?” can be answered with a certain degree of probability even without looking at the image itself. In the second example in Figure 7 of the appendix, the number 299 is written in large Arabic numerals, so even if MLLMs don't have the ability to read Italian from the image, they could probably answer the price as 299 euros. There are many examples like this that don't seem to be appropriate as VQA or multilingual language tasks, so the overall quality of the benchmark data is a concern.
  - Also, looking at Figure 5, it appears that words indicating native languages in each country's language (e.g. Deutsch in German, 日本語 in Japanese, Français in French, etc.) account for a certain percentage. In other words, there is a concern that many of the questions of this type, which ask about languages, will be answered correctly if the text is in French, for example, if the answer is Français.
- Importance as a benchmark
  - For example, we can evaluate the performance of multilingual QA using benchmarks such as Okapi [Lai+, EMNLP 2023], and we can evaluate the performance of QA that also includes text and asks about image understanding using benchmarks such as MMMU [Yue+, CVPR 2024]. The proposed benchmark dataset aims to satisfy these properties at the same time, but it is unclear what the performance and limitations of MLLM will be when it is used in this way.
  - Table 4 reports on the combination with OCR, but it is unclear what the accuracy of OCR is in the first place. It would also be necessary to evaluate the accuracy when the true value of the text contained in the image is given, rather than OCR.

**Questions:**

The reviewer would like to know the answer to the weaknesses above.

---

### Official Review · Reviewer_X52B · 2024-11-04

**Soundness:** 3
**Presentation:** 3
**Contribution:** 3
**Rating:** 6
**Confidence:** 3

**Summary:**

The paper proposes a VQA dataset where models must contextualize multilingual text in the image to answer questions. The paper provides extensive benchmarking and ablations on frontier open, closed models.

**Strengths:**

S1. Non-trivial data collection effort, which human annotators and quality control measures.

S2. Benchmarking of frontier models, both open and closed source, demonstrating gaps in current models. Extensive benchmarking in terms of number of models.

S3. Table 4 and experiments with OCR+text processing are both interesting and surprising. It also verifies that vision is critical to the benchmark, not just more simple OCR processing.

S4. Evidence that (multi-modal) few-shot prompting improves performance.

**Weaknesses:**

W1. It is not clear what Text-Centric Visual Question Answering is in the abstract. Given that this is a fairly specific task, it might be good to clearly define it or you risk your reader getting lost.

W2. Figure 4 could use some more explanation in the caption. As the figure stands now, it is not super clear how I should parse it or what the takeaways should be.

W3. L266, what OCR tool? Please cite.

W4. The benchmark frames languages as low resource; however, most languages are fairly wide spread.

W5. Can more recent open and closed source models be added (e.g., claude 3.5, llama 3.1, etc.)

**Questions:**

Please address the weaknesses above.

---

### Note · Authors · 2024-11-15

I have read and agree with the venue's withdrawal policy on behalf of myself and my co-authors.